# The Global Limb Anatomic Staging System (GLASS) for CLTI: Improving Inter-Observer Agreement

**DOI:** 10.3390/jcm10163454

**Published:** 2021-08-04

**Authors:** Joep G. J. Wijnand, Devin Zarkowsky, Bian Wu, Steven T. W. van Haelst, Evert-Jan P. A. Vonken, Thomas A. Sorrentino, Zachary Pallister, Jayer Chung, Joseph L. Mills, Martin Teraa, Marianne C. Verhaar, Gert J. de Borst, Michael S. Conte

**Affiliations:** 1Department of Vascular Surgery, University Medical Center Utrecht, 3584 CX Utrecht, The Netherlands; stevenvanhaelst@gmail.com (S.T.W.v.H.); M.Teraa@umcutrecht.nl (M.T.); G.J.deBorst-2@umcutrecht.nl (G.J.d.B.); 2Department of Vascular Surgery, UCSF Medical Center, San Francisco, CA 94143, USA; Devin.Zarkowsky@ucsf.edu (D.Z.); Bian.Wu@ucsf.edu (B.W.); thomas.sorrentino@ucsf.edu (T.A.S.); Michael.Conte2@ucsf.edu (M.S.C.); 3Department of Radiology, University Medical Center Utrecht, 3584 CX Utrecht, The Netherlands; e.vonken@umcutrecht.nl; 4Department of Vascular Surgery, Baylor College of Medicine, Houston, TX 77030, USA; zspallis@bcm.edu (Z.P.); Jayer.Chung@bcm.edu (J.C.); Joseph.Mills@bcm.edu (J.L.M.); 5Department of Nephrology & Hypertension, University Medical Center Utrecht, 3584 CX Utrecht, The Netherlands; m.c.verhaar@umcutrecht.nl

**Keywords:** CLTI, critical ischemia, GLASS, observer agreement

## Abstract

Objective: The 2020 Global Vascular Guidelines aim at improving decision making in Chronic Limb-Threatening Ischemia (CLTI) by providing a framework for evidence-based revascularization. Herein, the Global Limb Anatomic Staging System (GLASS) serves to estimate the chance of success and patency of arterial pathway revascularization based on the extent and distribution of the atherosclerotic lesions. We report the preliminary feasibility results and observer variability of the GLASS. GLASS is a part of the new global guideline and posed as a promising additional tool for EBR strategies to predict the success of lower extremity arterial revascularization. This study reports on the consistency of GLASS scoring to maximize inter-observer agreement and facilitate its application. Methods: GLASS separately scores the femoropopliteal (FP) and infrapopliteal (IP) segment based on stenosis severity, lesion length and the extent of calcification within the target artery pathway (TAP). In our stepwise approach, we used two angiographic datasets. Each following step was based on the lessons learned from the previous step. The primary outcome was inter-observer agreement measured as Cohen’s Kappa, scored by two (step 1 + 2) and four (step 3) blinded and experienced observers, respectively. Steps 1 (*n* = 139) and 2 (*n* = 50) were executed within a dataset of a Dutch interventional RCT in CLTI. Step 3 (*n* = 100) was performed in randomly selected all-comer CLTI patients from two vascular centers in the United States. Results: In step 1, kappa values were 0.346 (FP) and 0.180 (IP). In step 2, applied in the same dataset, the use of other experienced observers and a provided TAP, resulted in similar low kappa values 0.406 (FP) and 0.089 (IP). Subsequently, in step 3, the formation of an altered stepwise approach using component scoring, such as separate scoring of calcification and adding a ruler to the images resulted in kappa values increasing to 0.796 (FP) and 0.730 (IP). Conclusion: This retrospective GLASS validation study revealed low inter-observer agreement for unconditioned scoring. A stepwise component scoring provides acceptable agreement and a solid base for further prospective validation studies to investigate how GLASS relates to treatment outcomes.

## 1. Introduction

While Chronic Limb-Threatening Ischemia (CLTI) represents less than 10% of all PAD patients, it comes with a considerable burden in terms of morbidity, mortality and socio-economic costs. Despite improvement of the therapeutic armament, the amputation rate is up to 20% at twelve months [1,2], while over 50% of all CLTI patients die within 5 years after presentation [3,4].

Choices for revascularization are still not standardized and largely based on expert opinion and personal preference of the physician treating the patient [5]. Existing anatomic classifications in PAD are based on the location and severity of individual arterial lesions (e.g., TASC) [6] or quantify the overall burden of atherosclerotic disease [7]. These individual lesion-based classification systems correlate poorly with clinically effective revascularization in patients with CLTI and leave vascular specialists trying to integrate data for arterial segments into a management strategy for the whole limb.

The 2020 Global Vascular Guidelines aim at improving structured decision making in CLTI by providing a framework for evidence-based revascularization (EBR) [8]. This framework is composed of three dimensions: (1) Patient risk, (2) Limb status and (3) ANatomical pattern (PLAN). Components of each of those three dimensions are respectively: The Vascular Quality Initiative prediction model (VQI) for determining overall patient risk [9], WIfI for limb staging, and the Global Limb Anatomic Staging System (GLASS) for identifying different anatomical patterns of disease and related chance of success of revascularization. The writing group of the Global Vascular Guideline defined three GLASS stages based on the likelihood of immediate technical failure and one-year limb-based patency (LBP) following endovascular intervention of the selected TAP.

Although already incorporated in the guideline and suggested as a promising additional tool for EBR strategies to predict the success of lower extremity arterial revascularization, GLASS needs proper prospective validation. Our goal was to examine the consistency of GLASS scoring and to maximize inter-observer agreement to facilitate its application.

## 2. Methods and Results

### 2.1. GLASS Scoring Principles

The GLASS scoring principles have been reported previously in detail [8]. In short, GLASS staging requires separate scoring of the femoropopliteal (FP) and infrapopliteal (IP) segments (Table 1 (A,B)). Before doing so, the observer must identify the target artery pathway (TAP), which is the preferred IP artery for revascularization in the case (Figure 1). The TAP is defined by the proceduralist and thus identified either prospectively during a case, from operative notes, or based on imaging evidence of the IP artery that was primarily targeted for intervention. In the absence of such information, the least diseased IP artery on imaging is selected as the TAP. Furthermore, it is important to realize that GLASS was originally designed to be used for angiographic imaging. Hence, all the imaging and scoring that has been done in this study pertains to angiograms. 

Combinations of grade scores for the FP and IP segments are used to define three GLASS stages (Table 2) based on estimating the likelihood of immediate technical success and 12-month LBP, defined as maintained patency of the TAP following endovascular intervention. GLASS stages for the limb thus reflect a gradient of TAP complexity:

Stage I: Average Complexity Disease: technical failure <10% AND >70% 12-month LBP.

Stage II: Intermediate Complexity Disease: technical failure <20% AND 12-month LBP 50–70%.

Stage III: High Complexity Disease: technical failure >20%; OR <50% 12-month LBP.

For the present retrospective study, we used existing data sets to validate the GLASS. We report on the three stages that we run through in order to improve inter-observer agreement levels for GLASS scoring. The three steps were not predetermined. Each following step was the resulting effect of the outcome of the previous one. The study was approved by the institutional review board. The study was conducted in accordance with the declaration of Helsinki. All participants provided written informed consent.

### 2.2. Statistics

Cohen’s kappa values for variability were obtained using SPSS for Windows version 25.0 (SPSS Inc., Chicago, IL, USA). Kappa values of <0 reflects ‘poor’, 0 to 0.20 ‘slight’, 0.21 to 0.4 ‘fair’, 0.41 to 0.60 ‘moderate’, 0.61 to 0.8 ‘substantial’ and above 0.81 ‘almost perfect’ agreement. In addition to ordinal percentage agreement calculation, Kappa considers the possibility of the agreement occurring by chance. These statistics were applied in this and all subsequent steps.

#### 2.2.1. Step 1—Initial Inter-Observer Analysis within RCT Data

The inter-observer variability of the original GLASS was examined retrospectively in the Dutch multicenter PADI trial cohort [10] (Percutaneous Transluminal Angioplasty Versus Drug-Eluting Stents for Infrapopliteal Lesions in Critical Limb Ischemia), consisting of 139 patients eligible for predominantly infrapopliteal revascularization enrolled between 2007 and 2013. The study protocol, detailed patient characteristics and study results were previously published [9]. GLASS stage was determined based on the scoring system provided in Table 1 (A,B). Scoring was done by two experienced and independent radiologists. All 139 cases were used during step 1.

Results step 1

Kappa for FP was 0.346 (95% CI 0.126–0.566) and kappa for 0.180 (95% CI 0.078–0.282) (IP), fair and slight interobserver agreement, respectively. Evaluating the scoring process together with the observers, resulted in a strong belief that specific components in the scoring system, such as ‘severe calcification’, are more responsible for causing low agreement rates than others. Therefore, we decided that calcification kappa should be determined independently, resulting in a kappa value of 0.208 (95% CI −0.116–0.532) for FP calcification and 0.071 (95% CI −0.080–0.231) for IP calcification. It was also necessary to rule out certain potential factors contributing to variability such as differences in the observer-selected target artery pathway. Investigating the role of these factors led to the second step in our validation process.

#### 2.2.2. Step 2—Calibration Series within Dutch RCT

Step 2 consisted of a second scoring series by two vascular surgery fellows in the UCSF Medical Center (San Francisco, CA, USA), where diagnostic angiographies are incorporated in standard patient work-up and daily clinical practice. A total of 50 random cases from the same PADI cohort were scored using the same method as in step 1; however prior to scoring each limb, the TAP was predefined and provided to both independent observers.

Results step 2

Kappa for FP was 0.406 (95% CI 0.102–0.710) and kappa for IP was 0.089 (95% CI −0.007–0.171), again, fair and slight agreement, respectively. These findings confirmed that the high variability between observers was not caused by the background nor the TAP selection. Therefore, we aimed to identify potential contributing factors by splitting the original GLASS into a non-composite component scoring system, making it possible to assess each separate component within the GLASS scoring system.

#### 2.2.3. Step 3—Adjusted GLASS Inter-Observer Variability in US CLTI Cohort

In step 3 we adjusted the scoring routine (Table 3), based on the findings in steps 1 and 2, using component scoring to stimulate a more systematic approach. A total of 100 all-comer CLTI patients were randomly and retrospectively selected from two vascular centers in the United States, the Baylor College of Medicine (Houston, TX, USA) and the UCSF Medical Center (San Francisco, CA, USA). Each center provided 50 cases and two observers. Therefore, two separate 2 × 2 comparisons were done. To be able to trace back consistent discrepancies, we designed a roadmap for each segment by using only non-composite components scoring (Table 3). This approach enabled us to determine which component scores performed relatively better and worse in terms of observer agreement levels.

Results step 3—A

Almost all kappa’s for component scores increased drastically compared to the initial FP and IP combination scores. For instance, the total occlusion kappa value for the superficial femoral artery (SFA) and infrapopliteal (IP) trajectory reached almost 0.5. Contrarily, calcification alone showed much lower agreement levels (Table 4). In particular determining “severe calcification” in the IP trajectory caused high variability rates.

Results step 3—B

After consulting the observers in a “consensus session”, we found out that estimating length was another common factor amongst component scores causing high variability rates. This was confirmed in a sample of 20 randomly selected cases within the US cohort by adding a ruler to the images (for the purpose of measuring lesion length adequately); Kappa values increased to 0.796 (95% CI 0.656–0.936) (FP) and 0.730 (95% CI 0.61–0.85) (IP), representing substantial interrater agreement. (See Figure 2 for a total overview of the three steps).

## 3. Discussion

Through sequential optimization, we were able to obtain moderate interrater agreement for most non-composite (component) scores in GLASS. For the most part, the observer variability was attributable to the subjective aspect of estimating lesion length and severity of calcification. Although all of our observers were highly trained and experienced in reviewing angiographic imaging, high interobserver variability occurred for these components throughout the whole scoring process. Even though absolute accuracy in centimeter is not critical because the length-categories are estimates and described in proportions (e.g., 1/3 or 2/3 of length), when adding a ruler to the image interpretation, estimating lesion length became more reliable and reproducible, ultimately leading to ‘almost perfect’ agreement levels (kappa 0.8).

In comparison, existing classifications such as TASC show poor to moderate inter-observer agreement with kappa values ranging from 0.11 to 0.54 [11]. Moreover, individual lesion- or segment-based grading systems are less useful in daily practice where complex disease patterns are commonly encountered, especially in CLTI [11]. The clinical success of revascularization, particularly in patients with tissue loss, nearly always requires the restoration of direct arterial flow to the foot. In this regard, GLASS poses an important improvement over the lesion-based systems.

GLASS is a promising tool that can represent the “anatomical pattern” pillar in PLAN. When combined with tools for stratification of patient-risk and severity of limb threat, GLASS can facilitate the development of specific EBR strategies in CLTI. Furthermore, GLASS may enhance research quality in the future as it allows improved patient stratification, and thus, more homogeneous study populations can be formed, and better comparative studies can be executed to study the effect of specific endovascular interventions [12].

Future research should evolve around two main points: prospective validation and correlation with clinical presentation, treatment success and (limb-based) patency. Furthermore, the effects of pedal arch disease and patient factors on revascularization outcomes deserve attention in future research. Moreover, based on our results, it is important to search for a reproducible method to grade severity of calcification before its predictive value for infrainguinal interventions can be incorporated as factor within the GLASS scoring system. However, the system is meant for real world clinical use, and the current descriptor of “severe” is meant to be used whenever the treating specialist believes that the degree of calcification would significantly diminish the outcomes of an endovascular intervention in that segment. By design, this is subjective. There is no existing data to support any approach at present, except it is well established that severe calcification is indeed an important risk factor for both technical success and patency. Additionally, it would be interesting to see if and how artificial intelligence could play a role in an alternative approach in order to overcome described obstacles by implementing a computerized algorithm.

Our results show that accurate assessment of lesion length is essential to improve inter-observer agreement. Furthermore, severe calcification, particularly in the tibial arteries, is established as a negative predictor of technical success for interventions and portends higher amputation risk [13]. Considering the central role of CTA (and MRA) in diagnostic work-up in many countries, the ease of measuring lesion length with this imaging techniques and potential improved estimation and quantification of calcification underlines the need to also validate GLASS in CTA (and MRA) in future studies.

GLASS was designed for prospective use by clinicians who would define the preferred TAP based on the case at hand. In this regard it is likely that the assessment of lesion length, complete versus subtotal occlusion and even severity of calcification will be simpler to perform by the proceduralist in real-time. In contrast, our study applies directly to the use of GLASS for retrospective or registry-based research studies to better compare outcomes across stages of disease complexity. In these cases, use of a specified workflow as described herein and a core lab team of readers will help to insure the most consistent grading of disease patterns.

The main limitations of this study are the limited number of observers and the limited study size, especially in the second step. While fixing the TAP is a simplification of the GLASS and could be considered a limitation of our study; we experienced that in order to be able to record interobserver agreement, a fixed TAP was an essential requirement to obtain some level of standardization. Our preliminary validation data and optimization of the GLASS scoring system should be prospectively validated in a larger study population. Furthermore, given the new aspect of the scoring system, there might be a learning curve for the observers in the process. Part of the improvement may be contributed to this phenomenon. However, the observers from Houston showed similar agreement levels in step 3 during their first run compared to their UCSF counterparts who also served in the second step. It would be interesting to observe and explain how intra-observer agreement evolves once one becomes more familiar with the scoring system. Not including intra-observer agreement could also be considered as a limitation of this study.

## 4. Conclusions

We developed a stepwise approach for retrospective review of lower extremity angiograms using GLASS. Our systematic method improved the inter-observer agreement rates from an unacceptable low to an acceptable Kappa value. Further prospective validation studies using these workflows will determine the relationship between GLASS stage and treatment results in CLTI.

## Figures and Tables

**Figure 1 jcm-10-03454-f001:**
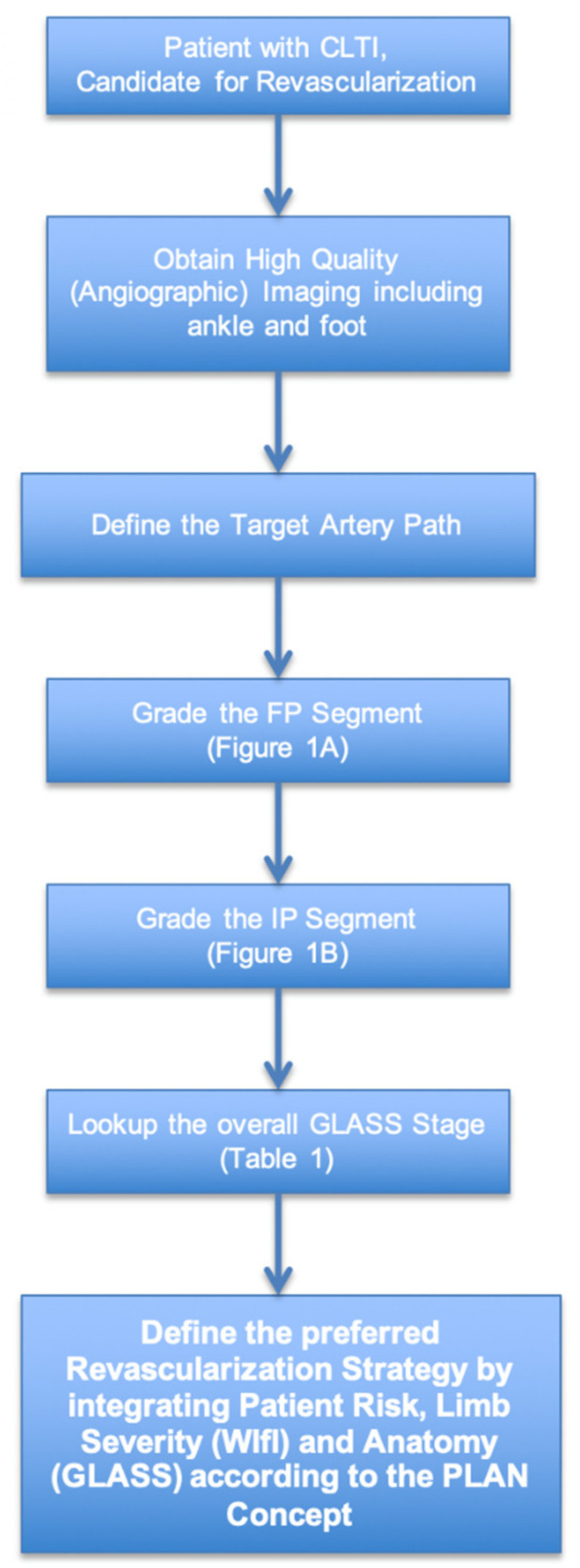
Flowchart illustrating use of GLASS for staging infrainguinal arterial disease.

**Figure 2 jcm-10-03454-f002:**
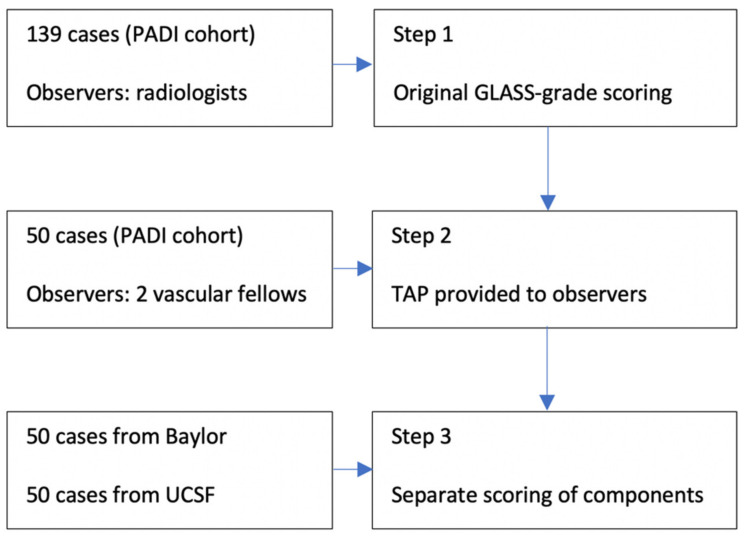
Methods workflow.

**Table 1 jcm-10-03454-t001:** (**A**) Original/Composite Femoropopliteal (FP) disease grading in GLASS. (**B**) Original/Composite Infrapopliteal (IP) disease grading in GLASS.

(A)	
Femoro-Popliteal (FP) Grading	
0	Mild or no significant (<50%) disease
1	Total length SFA disease <1/3 (<10 cm); may include single focal CTO (<5 cm) as long as not flush occlusion; popliteal artery with mild or no significant disease
2	Total length SFA disease 1/3–2/3 (10–20 cm); may include CTO totaling <1/3 (10 cm) but not flush occlusion; focal popliteal artery stenosis <2 cm, not involving trifurcation
3	Total length SFA disease >2/3 (>20 cm) length; may include any flush occlusion <20 cm or non-flush CTO 10–20 cm long; short popliteal stenosis 2–5 cm, not involving trifurcation
4	Total length SFA occlusion >20 cm; popliteal disease >5 cm or extending into trifurcation; any popliteal CTO
**(B)**	
**Infra-Popliteal (IP) Grading**	
0	Mild or no significant (<50%) disease
1	Focal stenosis <3 cm not including TP trunk
2	Total length of target artery disease <1/3 (<10 cm); single focal CTO (<3 cm not including TP trunk or target artery origin)
3	Total length of target artery disease 1/3–2/3 (10–20 cm); CTO 3–10 cm (may include target artery origin, but not TP trunk)
4	Total length of target artery disease >2/3 length; CTO >1/3 (>10 cm) of length (may include target artery origin); any CTO of TP trunk

A—Involvement of trifurcation means disease includes the origin of either the anterior tibial or tibioperoneal trunk. Severe calcification (e.g., >50% of circumference, diffuse, bulky or “coral reef” plaques) within the TAP increases the within-segment grade by +1. B—IP grading is applied only to the primary selected vessel in the TAP. Severe calcification (e.g., >50% of circumference, diffuse, bulky or “coral reef” plaques) within the TAP increases the within-segment grade by +1. TP trunk disease is only included if the TAP is the posterior tibial or peroneal artery.

**Table 2 jcm-10-03454-t002:** GLASS stages based on FP and IP grade.

	Infrainguinal GLASS Stage					
FP Grade	4	III	III	III	III	III
	3	II	II	II	III	III
	2	I	II	II	II	III
	1	I	I	II	II	III
	0	NA	I	I	II	III
		0	1	2	3	4
	IP Grade					

FP = femoropopliteal, IP = infra-popliteal.

**Table 3 jcm-10-03454-t003:** Altered non-composite scoring method.

SFA-Segment		Popliteal Segment	Infra-Popliteal Segment	
Total stenosis length	CTO	Total stenosis length	Total stenosis length	CTO
0 = mild or no significant (<50%) disease	0 = no CTO	0 = mild or no significant disease	0 = mild or no significant (<50%) disease	0 = no CTO
1 = <1/3 (<10 cm)	1 = <5 cm	1 = <2 cm	1 = <3 cm	1 = single focal CTO (<3 cm not including TP trunk or target artery origin)
2 = 1/3–2/3 (10–20 cm)	2 = <1/3 (10 cm)	2 = 2–5 cm	2 = < 1/3 (<10 cm)	2 = 3–10 cm (may include target artery origin, but not TP trunk)
3 = >2/3 (>20 cm)	3 = flush occlusion <20 cm	3 = >5 cm	3 = 1/3–2/3 (10–20 cm)	3 = >1/3 (>10 cm) of length (may include target artery origin)
	4 = non-flush CTO 10–20 cm long	4 = extending into trifurcation	4 = >2/3	4 = any CTO of TP trunk
	5 = SFA occlusion >20 cm	5 = popliteal CTO		

SFA = superficial femoral artery, CTO = chronic total occlusion.

**Table 4 jcm-10-03454-t004:** Kappa scores divided by component and center.

Component Score	UCSF 50	Baylor 50
SFA TSL	0.399 (0.195–0.603)	0.557 (0.398–0.736)
SFA CTO	0.498 (0.242–0.754)	0.486 (0.246–0.726)
FP Calc++	0.143 (−0.097–0.383)	0.483 (0.243–0.723)
Pop TSL	0.535 (0.354–0.716)	0.390 (0.236–0.544)
IP TSL	0.240 (0.062–0.418)	0.387 (0.217–0.557)
IP CTO	0.488 (0.324–0.652)	0.470 (0.289–0.651)
IP Calc++	−0.120 (−0.220–−0.020)	0.291 (−0.089–0.671)

SFA = superficial femoral artery, TSL = total stenosis length, CTO = chronic total occlusion, FP = femoropopliteal, IP = infra-popliteal (95% confidence interval).

## Data Availability

Not applicable.

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
