# Peer review of "The Global Limb Anatomic Staging System (GLASS) for CLTI: Improving Inter-Observer Agreement"

_jcm, 2021, doi:10.3390/jcm10163454_

Round 1

Reviewer 1 Report

Thank you for the opportunity to review. This manuscript presents a novel method for evaluating the validity of the new GLASS classification system for grading femoropopliteal and infrapopliteal atherosclerotic lesions. GLASS is included in the new Global Vascular Guidelines but has not yet been extensively validated. This study sought to identify inter-observer agreement, measured by Kappa statistics, between clinicians retrospectively scoring angiogram images. An iterative approach to the methods seems to have been used, with the study divided into three phases or steps, with changes made to the application and workflow of image interpretation between each step. These changes would then inform the methods of the next step to optimise scoring instructions and methods. Changes made included removal of calcification scores from the overall scoring, defining the target artery pathway prior to scoring, and the addition of a consistent measurement ruler to angiogram images.  The study concludes that moderate interobserver agreement could be reached with these changes to GLASS.

Some concerns exist regarding the clarity of methods and presentation of the results. Because of the iterative and evolving nature of the study methods (ie, changes were made to subsequent steps based on the results of the preceding steps), the methods and results of each step are presented sequentially. This is confusing and makes it difficult to evaluate the consistency and reasoning behind the methods. It would be helpful to either follow the standard separation between methods and results (preferred approach) or to have a summary statement about the steps that were taken and iterations of the methods. If the later option is chosen, results could be presented with descriptions about how the methods were altered based on preceding results.  

Steps are referred to in roman  (in text) and Arabic (in abstract) numerals. There is also inconsistency in the use of the terms describing the steps/phases of the study Eg line 133- recommend choosing one term (phase or step) and being consistent.

A description about what is meant by non-composite component scoring earlier in the methods would be helpful to readers as well as an explanation as to  why was this used in step II but not prior?

A statement about ethics is required.

Line 121- was this an impression (ie strong belief) or did you have quantification of the discrepancy between observer agreement was due to calcification? Please clarify if subsequent steps provided this quantification. 

Would the change in specialty between vascular fellow  (step II) and interventional radiologists (step I) introduce confounding/variability? How was this accounted for?

Line 140- sentence is a hung sentence- ?intended to be linked to previous sentence by a comma rather than period?

Lack of observer agreement with regard to calcification is important. This needs further exploration and discussion. The separation of calcification from the subsequent scoring algorithm questions whether this should be included in GLASS and if this is influential on ultimate GLASS scoring.

Line 232: ‘observe and explain’ rather than  ‘see and objectivate’?

Tables and Figures:

Table headings/legends would benefit from more detail to help reader understand table outside of context of article.

It would be helpful to have a table that shows the overall GLASS score (composite) kappa scores vs kappa scores for non-composite components for each step of the study. This would allow the reader to evaluate the merit of individual components vs the whole of GLASS.

References:

Is reference 12 published yet? Is there a DOI to quote?

Author Response

We would like to thank the reviewers for their constructive comments. Below you will find a point-by-point response to the reviewers’ comments and we adjusted the manuscript accordingly. All adjustments in the manuscript are trackable in the manuscript and depicted below. If reviewer comments did not result in manuscript changes we have motivated this in the responses.

What is also important to consider prior to reading the responses to reviewers’ comments, is that the goal of this study was an initial pilot to validate GLASS, identify sources of inconsistency and try to solve major issues in interobserver variance. The goal was not to provide extensive validation on large patient cohorts and compare GLASS performance in different patient populations. This is ideally done in larger prospective studies.

We used a pragmatic approach that provided us with direct available high quality angiographic images based on available datasets from the Netherlands and the USA. Although patient numbers and numbers of observers are relatively small, this study provides insights in potential sources of inconsistency in GLASS scoring and gives direction for future prospective validation studies. Therefore, this study provides relevant data for future GLASS validation studies and should be considered part of a sequence of validation and optimization of the GLASS classification system.

Reviewer 1

Thank you for the opportunity to review. This manuscript presents a novel method for evaluating the validity of the new GLASS classification system for grading femoropopliteal and infrapopliteal atherosclerotic lesions. GLASS is included in the new Global Vascular Guidelines but has not yet been extensively validated. This study sought to identify inter-observer agreement, measured by Kappa statistics, between clinicians retrospectively scoring angiogram images. An iterative approach to the methods seems to have been used, with the study divided into three phases or steps, with changes made to the application and workflow of image interpretation between each step. These changes would then inform the methods of the next step to optimise scoring instructions and methods. Changes made included removal of calcification scores from the overall scoring, defining the target artery pathway prior to scoring, and the addition of a consistent measurement ruler to angiogram images.  The study concludes that moderate interobserver agreement could be reached with these changes to GLASS.

Comment 1
Some concerns exist regarding the clarity of methods and presentation of the results. Because of the iterative and evolving nature of the study methods (ie, changes were made to subsequent steps based on the results of the preceding steps), the methods and results of each step are presented sequentially. This is confusing and makes it difficult to evaluate the consistency and reasoning behind the methods. It would be helpful to either follow the standard separation between methods and results (preferred approach) or to have a summary statement about the steps that were taken and iterations of the methods. If the later option is chosen, results could be presented with descriptions about how the methods were altered based on preceding results.  

We realize that the manuscript does not adhere to the usual Introduction, Methods, Results and Discussion pattern. But we decided to use a separate methods and results section for each step, since the method of each subsequent step resulted from the previous methods and obtained result in that specific step. In order to give more insight in the subsequential design and work-flow we added Figure 3 to the manuscript, see also response to comment 2 of reviewer 2.

Comment 2
Steps are referred to in roman  (in text) and Arabic (in abstract) numerals. There is also inconsistency in the use of the terms describing the steps/phases of the study Eg line 133- recommend choosing one term (phase or step) and being consistent.

We altered this so that it is consistent throughout the manuscript.

Comment 3
A description about what is meant by non-composite component scoring earlier in the methods would
be helpful to readers as well as an explanation as to  why was this used in step II but not prior?

Composite scores are described in figure 1A and 1B. Non-composite scores are described in table 2. In step 1 we applied GLASS as it was originally intended to be used. Throughout step 1 we noticed that we had to separate the different composite grades in order to be able to identify which specific aspects caused the low interobserver agreement rates.

Comment 4
A statement about ethics is required.

We added this; page 3 lines 101-103.

Comment 5
Line 121- was this an impression (ie strong belief) or did you have quantification of the discrepancy between observer agreement was due to calcification? Please clarify if subsequent steps provided this quantification. 

This observation was identified and quantified in early stages; that is why we looked at calcifications separately from the initial step (step I) (please see page 3 lines 125-127.) Calcification Kappa values remained low despite the alterations we made. (please see table 3).

Comment 6 
Would the change in specialty between vascular fellow  (step II) and interventional radiologists (step I) introduce confounding/variability? How was this accounted for?

We compared radiologists with radiologists and fellows with fellows. Furthermore, the GLASS system will ultimately be used by both, hence we have opted for a pragmatic approach, and the structured way to assess the images in this study should have reduced the differences to a minimum. For this study it was mandatory that all observers had extensive experience with scoring this kind of imaging.

Comment 7 
Line 140- sentence is a hung sentence- ?intended to be linked to previous sentence by a comma rather than period?

Thank you, we altered this.

Comment 8 
Lack of observer agreement with regard to calcification is important. This needs further exploration and discussion. The separation of calcification from the subsequent scoring algorithm questions whether this should be included in GLASS and if this is influential on ultimate GLASS scoring.

We agree. However, we think that we can only conclude that further research is necessary in order to investigate if and how calcification scoring should be implemented. Based on this pilot study we feel that such “recommendation” is too strong.

See the following lines:
“Moreover, based on our results it is important to search for a reproducible method to grade severity of calcification before its predictive value for infrainguinal interventions can be incorporated as factor within the GLASS scoring system. However, the system is meant for real world clinical use and the current descriptor of “severe” is meant to be used whenever the treating specialist believes that the degree of calcification would significantly diminish the outcomes of an endovascular intervention in that segment. By design, this is subjective.”

Comment 9
Line 232: ‘observe and explain’ rather than  ‘see and objectivate’?

We altered this.

Comment 10 
Tables and Figures:

Table headings/legends would benefit from more detail to help reader understand table outside of context of article.

We added some additional explanation.

Comment 11
It would be helpful to have a table that shows the overall GLASS score (composite) kappa scores vs kappa scores for non-composite components for each step of the study. This would allow the reader to evaluate the merit of individual components vs the whole of GLASS.

We named a few non-composite kappa values in our results section (page 3-4, lines 122-134);

“Kappa for FP was 0.346 [95% CI 0.126-0.566] and kappa for 0.180 [95% CI 0.078-0.282] (IP)” and
“Resulting in a kappa value of 0.208 [95% CI -0.116-0.532] for FP calcification and 0.071 [95% CI -0.080-0.231] for IP calcification”

We think that the abovementioned passage provides enough clarification and an extra table is not justified, but if the reviewer’s opinion is different we are happy to add an additional table.

Comment 12 
References:

Is reference 12 published yet? Is there a DOI to quote?

That is correct, we altered the citation.

Reviewer 2

Comment 1
For this work to be generalizable, I would recommend increasing the number of patients (angiograms) and the number of providers interpreting the angiograms.  In each step scoring was just done by 2 observers. 

Dear reviewer. Thank you for your valuable suggestion.
Since this is a pilot study with the aim to optimize the GLASS scoring we made the choice not to use a large number of observers and a larger number of angiograms. However, we agree with the reviewer that for adequate generalizability bigger series both in terms of cases as well as number of observers are desired in future prospective studies to further validate our findings. Furthermore, future studies should focus on how GLASS performs using different imaging techniques (i.e. CTA and MRA), which are for instance in most European countries easier accessible in sufficient numbers and sufficient quality. The number of high quality angiograms (not the preferred diagnostic method in the Netherlands) also limited our study population.

Comment 2
Workflow/methods is a little difficult to follow.  Would recommend a figure depicting work-flow

We agree that a work-flow figure clarifies the different steps/ stages throughout the process. We added such figure (see figure 3).

Comment 3
The step wise approach is a good method, but the authors do not address why there are 3 different datasets.  Why not share one universal dataset.  Having three different angiographic datasets may introduce bias as the patients may have different characteristics (i.e. more DM, ESRD in one data set).  If authors have to have multiple datasets I would recommend a table with patient demographics (this can be in the supplement)

We agree that the use of different datasets is by no means ideal in case you aim to relate scores to patient characteristics. However, in this study we wanted to investigate observer variability and reproducibility. We chose to use different datasets in order to minimize the impact of familiarity by observers with previously seen cases. We argue that even if there are statistical significant differences in baseline characteristics, this would not affect interobserver variability within those sets. However, it is indeed an interesting question of GLASS would perform different in for instance patient populations with a high grade of tibial artery disease compared to a population in which mainly the femoropopliteal segment is affected. This is something that should be evaluated in future studies, however this study was mainly focused on feasibility of the GLASS scoring method and methods to improve its interobserver agreement to facilitate further prospective studies to evaluate GLASS’s use and performance.

Reviewer 2 Report

For this work to be generalizable, I would recommend increasing the number of patients (angiograms) and the number of providers interpreting the angiograms.  In each step scoring was just done by 2 observers. 

Workflow/methods is a little difficult to follow.  Would recommend a figure depicting work-flow

The step wise approach is a good method, but the authors do not address why there are 3 different datasets.  Why not share one universal dataset.  Having three different angiographic datasets may introduce bias as the patients may have different characteristics (i.e. more DM, ESRD in one data set).  If authors have to have multiple datasets I would recommend a table with patient demographics (this can be in the supplement)

Author Response

(The authors gave the same response as above.)
